# Epidemiological Profile and Risk Factors for Malaria in Rural Communities Before the Operationalization of the Singrobo–Ahouaty Dam, Southern Côte d’Ivoire

**DOI:** 10.3390/tropicalmed10070197

**Published:** 2025-07-15

**Authors:** Taki Jean Deles Avenié, Kigbafori Dieudonné Silué, Négnorogo Guindo-Coulibaly, Naférima Koné, Sadikou Touré, Kouamé Laurent Valian, Kouassi Séraphin Kouadio, Alloua Marie Joelle Bédia, Boza Fulgence Déabo, Klotcholman Diabagaté, Christian Nsanzabana, Jean Tenena Coulibaly

**Affiliations:** 1Department of Medicine, University of Basel, Petersplatz 1, CH-4003 Basel, Switzerland; christian.nsanzabana@swisstph.ch (C.N.); 2Unité de Recherche et de Formation Bioscience, Université Félix Houphouët-Boigny, Abidjan 22 BP 770, Côte d’Ivoire; kigbafori.silue@gmail.com (K.D.S.); coulnegno1@yahoo.fr (N.G.-C.); konenaferima@gmail.com (N.K.); kouassiseraphin77@gmail.com (K.S.K.); allouabedia@gmail.com (A.M.J.B.); fulgencedeabo@gmail.com (B.F.D.); klotcholmand@gmail.com (K.D.); couljeanvae@yahoo.fr (J.T.C.); 3Centre Suisse de Recherches Scientifiques en Côte d’Ivoire, Abidjan 01 BP 1303, Côte d’Ivoire; sadikou.toure@csrs.ci; 4Swiss Tropical and Public Health Institute, CH-4123 Allschwil, Switzerland; 5Centre de Santé Urbain de Irobo, Jacqueville BP 201, Côte d’Ivoire; laurentvalian6@gmail.com

**Keywords:** malaria, prevalence, parasite density, risk factors, hydroelectric dam, Côte d’Ivoire

## Abstract

Malaria remains a major public health issue, especially near hydroelectric dams that often promote mosquito breeding. This study aimed to establish baseline epidemiological data during the construction of the Singrobo–Ahouaty dam to support assessment and decision-making for short- and long-term health impacts on surrounding communities. A cross-sectional survey was carried out in randomly selected households. Blood samples were analyzed using thick/thin smears and rapid diagnostic tests, while sociodemographic and behavioral data were collected via questionnaires. Statistical analyses included chi-square, Mann–Whitney, Kruskal–Wallis tests, and logistic regression. The malaria prevalence was 43.1% (394/915). The parasite density averaged 405.7 parasites/µL. School-age children (6–13 years) showed the highest prevalence (74.3%, *p* < 0.0001), while younger children (0–5 years) had the highest parasite density (1218.0 parasites/µL, *p* < 0.0001). Highly elevated infection rates (>51%) occurred in Sokrogbo, N’Dènou, and Amani-Menou, with the highest density in Ahérémou 1 (5663.9 parasites/µL). Risk factors included being an informal worker (ORa = 1.5), working in the raw material sector (ORa = 1.4) or market gardening/rice farming (ORa = 0.9; *p* = 0.043), and frequent mosquito bites (OR = 0.4; *p* = 0.017). These results underscore the need for stronger vector control strategies, improved bed net distribution and follow-up, and enhanced intersectoral collaboration in dam-influenced areas to reduce malaria transmission.

## 1. Background

Malaria remains a parasitic disease that continues to pose a significant public health threat worldwide, with the highest burden observed in sub-Saharan Africa [1,2,3,4]. The countries most affected are generally developing or low-income countries. In 2023, the estimated number of new malaria cases reached 263 million across 83 countries, with approximately 597,000 deaths attributed to the disease [2]. The African region of the World Health Organization (WHO) bears the highest malaria burden, accounting for 94% of global cases (over 247 million cases) and 95% of global malaria-related deaths (over 567,000 deaths) [2]. The highest mortality rates occur among children under five years of age [3,4]. Therefore, prompt and effective malaria treatment is particularly crucial for this age group. Malaria control efforts have primarily targeted children, as they represent the majority of reported cases [3,4] and remain the main target for control measures and research initiatives in areas of stable malaria transmission in sub-Saharan Africa [5]. In malaria-endemic regions, young children face a higher risk of progression to severe disease compared to adults [6]. They more frequently present with altered consciousness, respiratory distress, multiple seizures, severe anemia, hypoglycemia, acidosis, hyperlactatemia, and hyperparasitemia [6].

This disease remains one of the most widespread parasitic diseases in Côte d’Ivoire, significantly impacting public health and socioeconomic development [7,8]. In Côte d’Ivoire, malaria endemicity and epidemiology are closely linked to factors influencing the proliferation of *Anopheles* mosquito populations. These factors include rainfall, temperature, vegetation cover, and human activities that affect the environment [9,10,11,12]. *P. falciparum* remains the predominant species, accounting for more than 95% of malaria cases. However, other *Plasmodium* species, including *P. malariae*, *P. ovale*, and *P. vivax*, are also present in Côte d’Ivoire, responsible for less than 5% of malaria cases [13,14]. Despite ongoing control efforts by the government and its partners, malaria remains the deadliest endemic parasitic disease in the country [15]. In addition, there have been periods of high malaria transmission, with elevated prevalence rates, as observed in the Taabo department (the site of the present study), where malaria prevalence reached 46.0% in 2010 and 56.6% in 2011 [16]. This region is, therefore, a high-transmission area where a second hydroelectric dam, the Singrobo–Ahouaty dam, is currently under construction.

Hydroelectric infrastructure is vital in promoting irrigated agriculture, achieving food self-sufficiency, and fostering economic development in the West African region. The drive towards introducing this infrastructure has led several countries, including Côte d’Ivoire, to construct multiple hydroelectric dams [1]. Within this framework, Côte d’Ivoire has initiated the construction of the Singrobo–Ahouaty hydroelectric dam, with a capacity of 44 megawatts (MW). This dam is currently being built on the Bandama River in the Taabo district, specifically between the villages of Singrobo and Ahouaty, downstream of the Taabo dam [17].

Although hydroelectric dams provide socioeconomic and energy benefits, their environmental impacts can promote the spread of vector-borne diseases, including malaria, through the alteration of aquatic and environmental ecosystems. Moreover, the health aspect is often neglected in major water resource development projects [1,7,18]. Indeed, the construction of the Singrobo–Ahouaty dam could alter the local ecosystem and influence malaria transmission in the region. Hydroelectric dams have a major impact on the transmission of vector-borne diseases such as malaria, making them a public health issue [7,18,19,20]. Increased malaria transmission has been reported in Africa around several hydroelectric schemes [8,21,22]. Additionally, the socioenvironmental context around the dam is complex, characterized by demographic changes, environmental transformations, and population migratory movements associated with dam construction [17]. These factors can affect malaria transmission dynamics by altering mosquito vector habitats, water resource availability, and the living conditions of local populations.

However, there is a general lack of epidemiological data, particularly entomological data, in the study area, which is nevertheless recognized as a region with high malaria transmission [16]. In the context of the environmental changes induced by the construction of the Singrobo–Ahouaty dam, it is essential to collect both epidemiological and entomological data as well as malaria-related risk factors. This will serve as a basis for the short-, medium- and long-term assessments of the dam’s impacts on malaria transmission and the health of surrounding communities. The main objective of this study is to set up baseline epidemiological data on malaria transmission dynamics during the construction phase of the dam. A better understanding of this aspect is essential for decision-making and for the design and implementation of malaria prevention and control measures tailored to the specific needs of local communities. This could help reduce the morbidity and mortality associated with this parasitic disease in the context of water management.

## 2. Material and Methods

### 2.1. Study Area

A cross-sectional study was conducted in February 2021 across nine villages within the impact zone of the Singrobo–Ahouaty dam, which is currently under construction (Figure 1). These villages belong to the district of Taabo in the Agnéby–Tiassa region. The Taabo district (6°14′ North Latitude and 5°08′ West Longitude) is situated approximately 150 km from Abidjan, the economic capital of Côte d’Ivoire [16]. The Agnéby–Tiassa region is characterized by an extensive hydrographic network. The Bandama River, which hosts the Taabo hydroelectric dam, operational since the 1970s, serves as a crucial water resource. The resulting reservoir supports fishing activities, while the upstream area provides favorable conditions for food crop cultivation. The Singrobo–Ahouaty dam is being constructed downstream of the Taabo hydroelectric dam on the Bandama River. This water infrastructure project is led by Ivoire Hydro Energy (IHE), which aims to understand and monitor the transmission dynamics of water-related diseases during the dam’s construction phase. The villages included in the study were chosen based on their inclusion within the dam impact zone. Additionally, the Agnéby–Tiassa region, including the Taabo district, is recognized for its high malaria endemicity [16].

To this end, the Centre Suisse de Recherches Scientifiques en Côte d’Ivoire (CSRS) was commissioned to conduct this study within the dam’s impact zone. The selection of the study site was primarily driven by the ongoing construction of the Singrobo–Ahouaty dam. The region’s environmental conditions, including high humidity levels and extensive areas, create permanent reproduction sites for malaria vectors (*Anopheles gambiae*, *An. Funestus,* and *An. nili*) such as malaria parasites (*P. falciparum*, *P. malariae*, and *P. ovale*) [16]. These conditions are further exacerbated by the substantial annual rainfall, which reached 1398 and 1544 mm in 2020 and 2021, respectively, with an average of 1332 mm over the period 2012–2021. The average temperature of the study area was recorded at 26.2 °C in 2020 and 26.1 °C in 2021, with a mean of 26.1 °C over the same period (2012–2021, according to data from NASA POWER) (Figure 2).

### 2.2. Study Design

After obtaining ethical approval from Comité National d’Ethique des Sciences de la Vie et de la Santé en Côte d’Ivoire (CNESVS) (protocol code: 149-22/MSHPCMU/CNESVS-kp), local authorities, parents, and/or legal guardians of preschool and school-age children were informed about the objectives, procedures, risks, and benefits of the study. Community health workers (CHWs) received training from the research team on their role in conducting the study. This study was carried out through a cross-sectional parasitological survey. Capillary blood samples were collected in the nine villages within the impact zone of the Singrobo–Ahouaty dam to diagnose malaria using rapid diagnostic tests (RDTs) and mixed smears (thick drop and blood smear). The RDTs used were based on Histidine-Rich Protein 2 (HRP2). In each studied village, households were randomly selected using a geographic information system (GIS). This system, containing the geographical coordinates of the selected households, was then integrated into the Android application “OsmAnd–Maps & Navigation”, installed on the smartphones of all survey team members. A specific training session on the use of this system and the retrieval of household coordinates was provided to the entire team by its developer. Once in the field, households were located and identified using GIS-equipped smartphones. Upon arrival at a household, the researchers and the guide (fluent in the local language) explained the study objectives, procedures, risks, and benefits. Individuals who agreed to participate provided informed consent by signing a consent form (≥18 years), while minors (<18 years) provided assent before a capillary blood sample was collected from their fingertip. The RDTs and mixed smears were conducted on-site in the various households. The analysis of the RDTs and mixed smears helped identify the epidemiological profile of malaria in the study area. Additionally, questionnaires were administered to household heads in homes where malaria diagnoses were performed to assess the risk factors associated with malaria.

The questionnaires were developed and validated by the project’s investigators. Other academic researchers involved in the management of the project also validated the questions prior to conducting the interviews. Thus, all nine localities in the study area were investigated. The inclusion criteria were agreeing to participate by signing an informed consent form and/or assent if applicable, residing in one of the villages within the study area, agreeing to provide a capillary blood sample, and being asymptomatic. The non-inclusion criteria included lack of informed consent from the participant or parent/guardian for minors (under 18 years), presenting symptoms and/or signs of severe, chronic, or central nervous system infections, patients presenting signs of severe malaria, and confirmed or suspected pregnancy. Participation in this study was voluntary. The questionnaires were addressed to the household’s head and were designed to assess malaria-associated risk factors.

### 2.3. Sample Size Calculation

This study forms part of the health component of the Singrobo–Ahouaty Hydroelectric Development Project, which aims to tackle several water-borne diseases, including malaria. The number of households required for the implementation of the project to which this study is linked was estimated. For the purposes of this study, a household is defined as a group of individuals living under the same roof and sharing the same domestic economy. A 5% variation in indicators from one year to the next during the three-year dam construction period was considered significant. Assuming that 28% of households in the project area had an acceptable health status with regard to the targeted diseases, the objective was to increase this proportion by 15 percentage points each year, with the aim of achieving good health for 75% of households by the end of the three-year period. This estimate was made with a 95% confidence level and 80% statistical power. The number of households required to detect such a change was calculated as follows:*n* = D [(Zα + Zβ)^2^ × (P1(1 − P1) + P2(1 − P2))/(P2 − P1)^2^]. So,*n* = 2[(1.645 + 1.282)^2^ × (0.28(1 − 0.28) + 0.75(1 − 0.75))/(0.75 − 0.28)^2^] = 30 households/village,
where *n* = number of households, D = design effect, conditionally estimated at 2, P1 = estimated proportion of households in the project area with an acceptable health status, P2 = expected proportion of households with improved health status at the end of the project, Zα = Z-score corresponding to the confidence level required to conclude that a change has occurred and is not due to chance, and Zβ = Z-score corresponding to the confidence level required to detect the change if it actually occurs. Thus, 30 households per village were selected for study, making a total of 270 households across the nine villages within the impact zone of the Singrobo–Ahouaty dam project. However, for this malaria-specific study, 252 households were randomly selected with a proportional distribution according to the number of households in each village within the zone of influence of the Singrobo–Ahouaty dam. From each household, four individuals were selected: a parent (aged 18 or over, either the father or mother), an adolescent (aged 14–17), a school-age child (aged 6–13) and a preschool child (aged 5 or under). However, certain age groups, such as teenagers and school- or preschool-age children, were absent from several households. In addition, several cases of refusal were recorded at both the household level and among individuals within households.

### 2.4. Blood Sample Collection and Analysis

Finger prick blood samples and body temperatures were collected from the household’s members. Before the data and sample collection process, the study was explained to participants, and they signed an informed consent or assent form for minors (<18 years). Malaria diagnosis was performed using rapid diagnostic tests (RDTs) and thick and thin blood smears (TBSs). The TBSs were then prepared on microscopy slides and subsequently transported to the laboratory for adequate preparation and readings.

❖Procedures and Analysis of TDR (HRP2)

After disinfecting the participant’s fingertip with an alcohol swab, a sterile lancet was used to collect a drop of blood, which was then placed on the test cassette using a capillary loop. The participant’s finger was carefully disinfected again with an alcohol-soaked cotton pad. Subsequently, 2 to 3 drops of buffer solution were added according to the manufacturer’s instructions, and the test was left to incubate for 10 to 15 min. The result was read visually: the presence of two lines indicated a positive test, a single control line signified a negative test, and the absence of a control line rendered the test invalid. Finally, all used materials were disposed of in a biomedical waste container, and hands were thoroughly washed.

❖Laboratory Procedures and Analysis of TBSs under the Optical Microscope

First, a 10% solution of Giemsa was prepared and left in a container. Next, thin blood smears were fixed with methanol using a Pasteur pipette and air-dried on a flat surface for three minutes. All TBSs were then stained with Giemsa 10% for 20 min and then rinsed with distilled water (pH = 7) [23]. Finally, the TBSs were air-dried before being examined under a light microscope (×100) with immersion oil. The TBSs were analyzed by two qualified technicians, with each slide examined only once. A quality control check of the microscopy readings was performed by a senior technician for 10% of randomly selected slides. Specifically, 5% of the slides analyzed by each technician were reassigned to the other for quality control. The quality control results for the TBSs were largely consistent with the individual results obtained by each technician.

### 2.5. Identification of Malaria Risk Factors

The identification of risk factors was conducted through the questionnaire survey targeting heads of households. Interviews were conducted individually. Each household’s head or representative was invited to respond to questions about their quality of life, malaria-related behaviors, and other relevant information such as socioeconomic, demographic, environmental, and behavioral factors and knowledge and attitude in relation to malaria. Participants responding to the questionnaire were previously reassured that the confidentiality of their responses would be fully respected.

### 2.6. Statistical Analysis

The data were double-entered into Excel version 2019. Both files were then merged into a single database according to the data source. This database was exported into R version 4.3.2 software and RStudio version 2024.12.1-563 to calculate the prevalence and parasite density of malaria, expressed in proportions and geometric means, respectively. Odds ratio (OR) tests were used to evaluate the infection risk between genders, Pearson’s chi-square tests were used to compare proportions, and the Wilcoxon test was used to compare the parasite density between genders. Kruskal–Wallis tests were used to compare the parasite density between age groups. Associations between parasitic infections and questionnaires were assessed via binary logistic regression (BLR), with a significance threshold of 5%.

## 3. Results

### 3.1. Characteristics of the Study Participants

This study included 915 individuals aged from three months to 91 years old living in the nine villages of the study area. The majority of participants were female (*n* = 541; 59.1%, 95% CI: 55.9–62.4), and they were older (29.5 years) compared to the males (25.5 years) (Z = −3.108; *p* = 0.002). Most participants were aged ≥14 years (*n* = 567, 62.0%; 95% CI: 58.9–65.2), compared to those aged 6 to 13 years (*n* = 140; 15.3%, 95% CI: 13.1–17.5) and those aged 0 to 5 years (*n* = 208; 22.7%, 95% CI: 20.0–25.5).

### 3.2. Malaria Infection Prevalence

Table 1 and Table 2 present the prevalence of malaria infection by gender and age group in different villages within the impact zone of the Singrobo–Ahouaty hydroelectric dam in 2021. The malaria prevalence was assessed by combining the results of the RDTs and TBSs (microscopy). An individual was considered malaria-positive when (i) the RDT alone was positive, (ii) the TBS alone was positive, or (iii) both the RDT and TBS were positive. The analysis was based on the odds ratio (OR) test, with “Male” as the reference category. The overall malaria prevalence in the study area was 43.1%, with 394 infected individuals out of 915 diagnosed. Rapid diagnostic tests (RDTs) detected 330/915 cases (36.1%), and TBSs detected 232/875 cases (26.5%). The prevalence was higher with RDTs compared to microscopy (χ^2^ = 169.738; *p* < 0.0001) (Table 1). Furthermore, asymptomatic malaria [95.9% (16/386), CI95%: 93.7–97.7] was more prevalent than symptomatic malaria [4.1% (370/386), CI95%: 2.3–6.3], which is defined as having a fever of at least 37.5 °C (Figure 3).

❖Malaria Infection Prevalence by Gender

Overall, men had a slightly higher prevalence (46.8%) than women (40.5%), but this difference was not statistically significant (*p* = 0.067), although it was close to the 0.05 threshold. The OR (1.3) suggests that men were approximately 1.3 times more likely to be infected than women, but the confidence interval (1.0–1.7) includes 1, indicating that this difference could be due to chance. No statistically significant differences were observed between men and women in any of the villages (*p* > 0.05 in all cases). N’Dènou (60.0%; OR = 1.7), Amani-Menou (59.0%; OR = 1.5), and Pacobo (37.5%; OR = 2.6) showed higher malaria prevalence among men, but these differences were not statistically significant (*p* > 0.05). In contrast, in Ahérémou-1 (48.3%; OR < 1), the prevalence was slightly higher among women, but again, the difference was not significant (*p* > 0.05) (Table 1).

❖Malaria Infection Prevalence by Age Groups

The malaria prevalence was 30.2% (171/567) among adolescents and adults (≥14 years), 74.3% (104/140) among school-age children (6–13 years), and 57.2% (119/208) among preschool children (0–5 years). The differences between the age groups were statistically significant (χ^2^ = 111.155; *p* < 0.0001) (Table 2). This difference between age groups was significant for Ahérémou-1, Singrobo, Ahouaty, Sokrogbo, Ahérémou-2, N’Dènou, and Amani-Menou (*p* < 0.05), but not in Pacobo and Kotiessou. In most villages, malaria prevalence was highest among school-age children, except in Singrobo, where the highest prevalence was observed in preschool children (Table 2).

❖Malaria Infection Prevalence by Village

The malaria prevalence varied significantly between villages (χ^2^ = 34.653; *p* < 0.0001) and was recorded as follows: 43.9% (25/57) in Ahérémou-1, 38.5% (42/109) in Singrobo, 26.6% (25/94) in Pacobo, 32.1% (27/84) in Ahouaty, 55.0% (83/151) in Sokrogbo, 35.9% (46/128) in Ahérémou-2, 41.9% (26/62) in Kotiessou, 51.2% (42/82) in N’Dènou, and 52.7% (78/148) in Amani-Menou. The highest prevalence was observed in Sokrogbo, N’Dènou, and Amani-Menou (Table 1). There was also no significant difference in the prevalence of asymptomatic and symptomatic malaria between the villages, although the rate of symptomatic malaria was slightly higher in Kotiessou (χ^2^ = 9041; *p* = 0.339) (Figure 3).

❖Proportion of Infection with *Plasmodium* spp.

A total of 915 RDTs were performed, of which 875 were also analyzed using TBS. The RDTs proved to be more sensitive than the TBS, as 318 samples tested positive with RDTs, while only 232 were detected as positive by TBS. Microscopy was used to identify *P. falciparum* and *P. malariae*. *P. falciparum* was the predominant species, detected in 228 individuals (98.3%), while *P. malariae* was detected in only three cases (1.7%), including a 56-year-old man in Pacobo and two preschool children in Sokrogbo. Additionally, a single case of mixed infection (0.4%) with both *P. falciparum* and *P. malariae* was observed in a 5-year-old boy in the village of Sokrogbo.

### 3.3. Malaria Parasite Density

Table 1 and Table 2 present the geometric mean parasite densities across the nine villages, stratified by gender and age groups. The parasite-negative individuals were removed from the parasite density estimates. Among the 915 blood samples collected, 875 were analyzed using light microscopy, of which 232 tested positive. The overall parasite density was 405.7 parasites/µL.

❖Malaria Parasite Density by Gender

On average, the geometric mean densities were 359.8 parasites/µL in women and 475.3 parasites/µL in men, with no significant difference between the two groups (Z = −1.528, *p* = 0.127). Similarly, across all villages, no significant association was observed between parasite density and gender (*p* > 0.05), except in Ahérémou-1, where men exhibited a moderately higher parasite density of 1208.2 parasites/µL (Z = −2.521, *p* = 0.012). In Amani-Menou, although the parasite density was higher in men (608.3 parasites/µL) than in women (289.8 parasites/µL), the difference was not statistically significant (Z = −1.909, *p* = 0.056) (Table 1).

❖Malaria Parasite Load by Age Groups

Overall, the parasite density was 160.4 parasites/µL in adolescents and adults (≥14 years), 509.0 parasites/µL in school-age children (6–13 years), and 1219.0 parasites/µL in children aged 0–5 years (ꭓ^2^_kw_ = 43.383, *p* < 0.0001). The parasite density was relatively similar across the three groups in all villages (*p* > 0.005), except Sokrogbo (ꭓ^2^_kw_ = 12.223, *p* = 0.002) and Amani-Menou (ꭓ^2^_kw_ = 18.242, *p* < 0.0001), where children aged 0–5 years exhibited moderate parasite density (Table 2).

❖Parasite Density by Village

The geometric means of parasites densities of malaria in the nine villages were 5663.8 parasites/µL, 305.0 parasites/µL, 860.2 parasites/µL, 363.3 parasites/µL, 566.3 parasites/µL, 133.6 parasites/µL, 242.1 parasites/µL, 181.2 parasites/µL, and 409.3 parasites/µL, respectively, in 2021, in Ahérémou 1, Singrobo, Pacobo, Ahouaty, Sokrogbo, Ahérémou-2, Kotiessou, N’Dènou, and Amani–Menou (Table 1). A significant difference (ꭓ^2^_kw_ = 31.080; *p* < 0.0001) was observed between the parasite density in different villages, and Ahérémou 1 had the highest parasite density.

### 3.4. Malaria Risk Factors

In this study, 244 household heads from different households were surveyed using a 40-question questionnaire covering socioeconomic, demographic, environmental, and behavioral characteristics, as well as malaria-related knowledge. The aim was to assess potential associations between malaria infection and these factors using binary logistic regression (BLR). BLR was applied to all 40 independent variables (Appendix A). Variables with a significant *p*-value (*p* < 0.05) were reported in another table and considered risk factors for malaria transmission. Adjusted odds ratios (ORa) were then calculated. The analyses revealed three main risk factors for malaria transmission in the study area: occupation, monthly income, and the frequency of nuisance due to mosquito bites. In this context, mosquito nuisance refers to residents’ perception of how frequently they experience mosquito bites in their homes (Table 3).

The analyses show that occupation appears to be a determining factor in malaria transmission. Compared to unemployed people, informal sector workers had a significantly higher risk of malaria infection (ORa = 1.5; *p* = 0.017) (Table 3).

Similar increases in risk were observed among raw materials workers (ORa = 1.4; *p* = 0.048) and market gardeners (ORa = 0.9; *p* = 0.043). However, the latter group experienced a protective effect. Monthly income is also a factor associated with malaria transmission, with surprising results. Compared to individuals earning FCFA 60,000 or more, those earning less than FCFA 60,000 are at a greater risk of malaria infection (ORa = 0.7; *p* = 0.037). This is a statistically significant result, given that lower income is generally associated with living conditions that favor malaria transmission. No significant association was found for individuals with no income (*p* = 0.216). Mosquito nuisance frequency is strongly associated with the risk of malaria infection. Those reporting infrequent mosquito nuisance were less likely to contract the disease than those exposed several times a day (ORa = 0.4; *p* = 0.017) (Table 3).

## 4. Discussion

This study examined the epidemiological profile and risk factors of malaria around the Singrobo–Ahouaty hydroelectric dam, which is currently under construction. The aim was to gain a comprehensive understanding of malaria epidemiology and key transmission determinants during the construction phase, with the ultimate aim of assessing how short-, medium-, and long-term environmental and socioeconomic changes induced by the construction and operation of the dam may affect the malaria prevalence and transmission in the surrounding villages.

In this study, the overall prevalence of malaria around the Singrobo–Ahouaty dam was high (43.1%). Although this prevalence rate is approximately equal to 46.0% and lower than the 56.6% reported in studies conducted in 2010 and 2011, respectively [16], it is also approximately comparable to the 40.8% observed in a recent study conducted in south-eastern Côte d’Ivoire [24]. This difference may be due to the fact that the 2010–2011 study was conducted during the rainy season (June–July) [16], whereas the current study was conducted during the dry season (February 2021). In addition, studies in Cameroon (15%) and Tanzania (13%) reported lower prevalence rates [25,26], probably due to country-specific factors influencing malaria transmission. Overall, men and women were exposed to a similar risk of malaria infection. Behavioral and occupational patterns specific to each group result in similar levels of exposure. Common risk factors include prolonged outdoor exposure to mosquito bites and water-related agricultural activities such as market gardening and rice cultivation [27]. A previous study also reported higher malaria prevalence among men [16]. School-aged children (6–13 years) were more susceptible to malaria infection than younger children (0–5 years) and adolescents/adults (≥14 years). This finding is consistent with a study from Cameroon, which found that children aged 60–119 months were more likely to be infected with malaria than their younger siblings [26]. Older children are more exposed due to behaviors that increase their risk of mosquito bites [28,29,30], such as playing outdoors at night or sleeping without protective clothing or bed nets. In addition, their immunity to malaria is still developing, making them more susceptible to infections than adolescents and adults (≥14 years). Of the villages surveyed, Sokrogbo, N’Dènou, and Amani-Menou had the highest malaria prevalence. Surrounded by lowlands used for rice cultivation and market gardening, these villages have permanent water resources, even during the dry season, which favor mosquito breeding and contribute to malaria transmission [14,31,32]. In this study, *P. falciparum* and *P. malariae* were the only species detected, with *P. falciparum* accounting for 98.3% of cases. The absence of *P. ovale* may be due to its low prevalence [14,33,34,35,36] or the focus of the study on villages, unlike previous studies that included both rural and urban areas [16]. Asymptomatic malaria is widespread in all the villages in the study area, whereas symptomatic malaria appears to be slightly more prevalent in Kotiessou. This may be because current malaria control strategies do not consider asymptomatic cases, which continue to circulate undetected in the population [37]. Furthermore, Kotiessou is located close to the Tabbo hydroelectric dam, whose reservoir could create favorable conditions for the proliferation of malaria-carrying mosquitoes, thereby increasing the population’s exposure to infectious bites. Across all villages except Ahérémou 1, children aged 0–5 years had the highest parasite density (33,747.5 parasites/µL), reflecting the gradual acquisition of immunity to *Plasmodium* [16,38,39]. In endemic regions, young children initially have low immunity, making them more susceptible to infection, which improves with age [16,27]. The remarkably moderate parasite density in Ahérémou 1 (5663.9 parasites/µL) is mainly due to marked age-related contrast, with significantly high values observed in children (33,747.5 parasites/µL). This phenomenon is probably due to the relative weakness of their immunity to *Plasmodium* infection [38,39]. Overall, in highly endemic regions, children under five years of age tend to have higher infection rates than older children, adolescents, and adults because their host–parasite interactions remain unstable, allowing for higher parasite densities without typical malaria symptoms [40]. In Sokrogbo, men had significantly higher parasite densities, possibly related to occupational activities that increase their exposure to mosquitoes. This study highlights a marked heterogeneity of malaria transmission across villages and age groups and underscores the need for targeted malaria control strategies that take into account local variation, especially among young children. In this study, RDTs detected more positive cases than microscopy because they identify parasite antigens such as HRP2, which can persist after live parasites have been cleared, leading to false positives, especially in recently treated individuals [41]. HRP2-based RDTs can remain positive for 2–4 weeks, overestimating malaria prevalence [41]. In contrast, microscopy detects only active parasites [42]. Furthermore, in submicroscopic infections (parasitemia < 50 parasites/µL), RDTs remain positive, while microscopy fails [43].

Occupation is one of the factors that determines malaria transmission. Compared with unemployed people, workers in the informal sector are at a significantly higher risk of malaria infection. These results align with several other recent studies conducted in sub-Saharan Africa [44,45,46]. This could be due to working conditions exposing these individuals to environments conducive to mosquito proliferation, such as swampy areas, precarious housing, and limited access to preventive measures. An increased risk has also been observed among workers in the raw materials industry (cocoa, rubber, and oil palm). These workers are often required to work in forested areas or near stagnant water, which may increase their contact with malaria vectors. As with commodity workers, market gardeners and rice growers are also at an increased risk of malaria infection. Studies conducted in central and northern Côte d’Ivoire have emphasized the significant contribution of these agricultural activities to malaria transmission [47,48]. This could be explained by the nature of these activities, which are often carried out outdoors in areas close to wetlands or stagnant water, which are conducive to the proliferation of malaria vectors, particularly *An. gambiae* s.l. [49,50], which is the main malaria vector in the study area. This would make this group more susceptible to being bitten by malaria-carrying mosquitoes. These results emphasize the importance of considering occupational conditions when devising malaria control strategies, particularly when targeting high-risk groups such as informal workers and agricultural workers. Monthly income also appears to be a factor associated with malaria transmission. Indeed, individuals with a monthly income of less than 60,000 XOF have a significantly higher risk of malaria infection than those with an income equal to or above this threshold. Several other recent studies carried out in Côte d’Ivoire have also observed this phenomenon [45,51,52]. This highlights the impact of socioeconomic factors on vulnerability to the disease. Households with limited financial resources often face precarious living conditions characterized by inadequate housing that offers little protection against mosquitoes (e.g., absence of mosquito nets or open dwellings), proximity to mosquito breeding grounds (e.g., stagnant water), and limited access to preventive measures or healthcare in the event of fever. Furthermore, health-related expenses can pose a significant financial burden on low-income families, delaying access to treatment and perpetuating transmission foci. This finding underlines the need to integrate a socioeconomic approach into malaria control policies. In addition to socioeconomic factors, increased exposure to mosquito bites has been identified as a determining factor in the risk of malaria infection. This finding is consistent with vector biology and malaria transmission dynamics, both of which rely on infected *Anopheles* mosquitoes biting humans [43,53,54,55]. Frequent exposure can result from various factors, such as the absence or non-use of long-lasting insecticidal treated nets (LLINs), engaging in outdoor activities in the evening or at night, wearing unsuitable clothing, or living in poorly protected housing (e.g., windows without screens or open roofs). In Taabo, where climatic and environmental conditions favor mosquito proliferation, such exposure constitutes a major risk. This observation underlines the importance of incorporating vector prevention into control strategies, such as the mass distribution of LLINs, educating people on how to use them properly, improving housing, and running awareness campaigns about risky behavior.

After the operationalization of the Singrobo–Ahouaty dam, the permanent water body and suitable ecological conditions, combined with seasonal variations in rainfall, could further increase the risk of transmission. Malaria is rainfall-driven, and high rainfall is likely to enhance the survival and reproduction of vectors, consequently increasing the risk of malaria transmission [56].

### Limitations of the Study

This study is part of the “Singrobo-Ahouaty Hydroelectric Development Project” health component, which targets several waterborne diseases, including malaria. However, the molecular aspect is not included in this project, preventing the implementation of molecular malaria diagnosis and the assessment of the genetic diversity of *P. falciparum* populations in the study area. The lack of molecular methods has limited malaria assessment to conventional methods such as RDTs and microscopy. PCR could have detected infections with low parasite loads and assessed genetic diversity, which is crucial for understanding transmission dynamics and the emergence of resistance.

## 5. Conclusions

This study examined the epidemiological profile and risk factors for malaria around the Singrobo–Ahouaty hydroelectric dam, highlighting a high prevalence. Men and women are equally at risk of malaria infection. Furthermore, school-age children are more exposed. *P. falciparum* was the most dominant species, while *P. ovale* was absent in this study. The key risk factors implicated in malaria transmission in the study area are occupation, monthly income, and an increased frequency of mosquito bites. It is vital to strengthen malaria prevention strategies, particularly through the use of LLINs. Regular maintenance ensures their effectiveness, reduces malaria transmission, and prevents mosquitoes from becoming resistant to insecticides. We recommend that health authorities and the WHO prioritize the distribution of good-quality nets, especially those adapted to the context, monitor their use, and launch awareness campaigns. This study argues for a community-based approach to malaria control in areas affected by development projects. Multi-sectoral collaboration between health, environment, and energy stakeholders is essential to integrate malaria control into dam management and to minimize health impacts on local communities. Continuous monitoring and further research are essential to track mosquito population dynamics and sporozooïtes and entomological inoculation infection rates and to adapt control strategies in this region, which will also benefit from a second hydroelectric dam.

## Figures and Tables

**Figure 1 tropicalmed-10-00197-f001:**
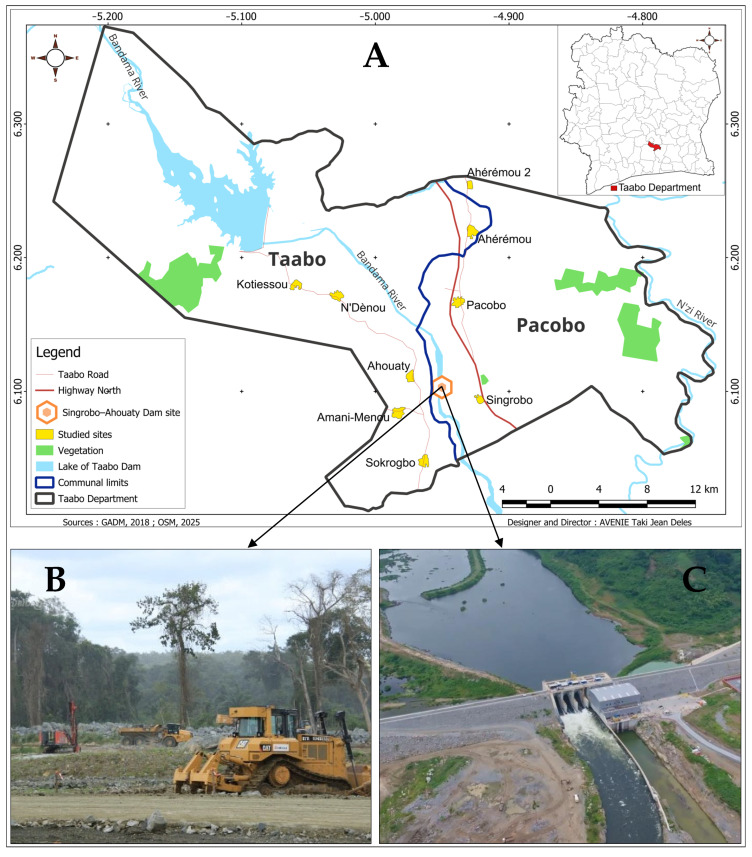
Map of the study area. (**A**): Geographic location of rural communities living in the area of the Singrobo–Ahouaty dam; (**B**,**C**): status of the construction progress of the Singrobo–Ahouaty dam in 2021 and 2024, respectively.

**Figure 2 tropicalmed-10-00197-f002:**
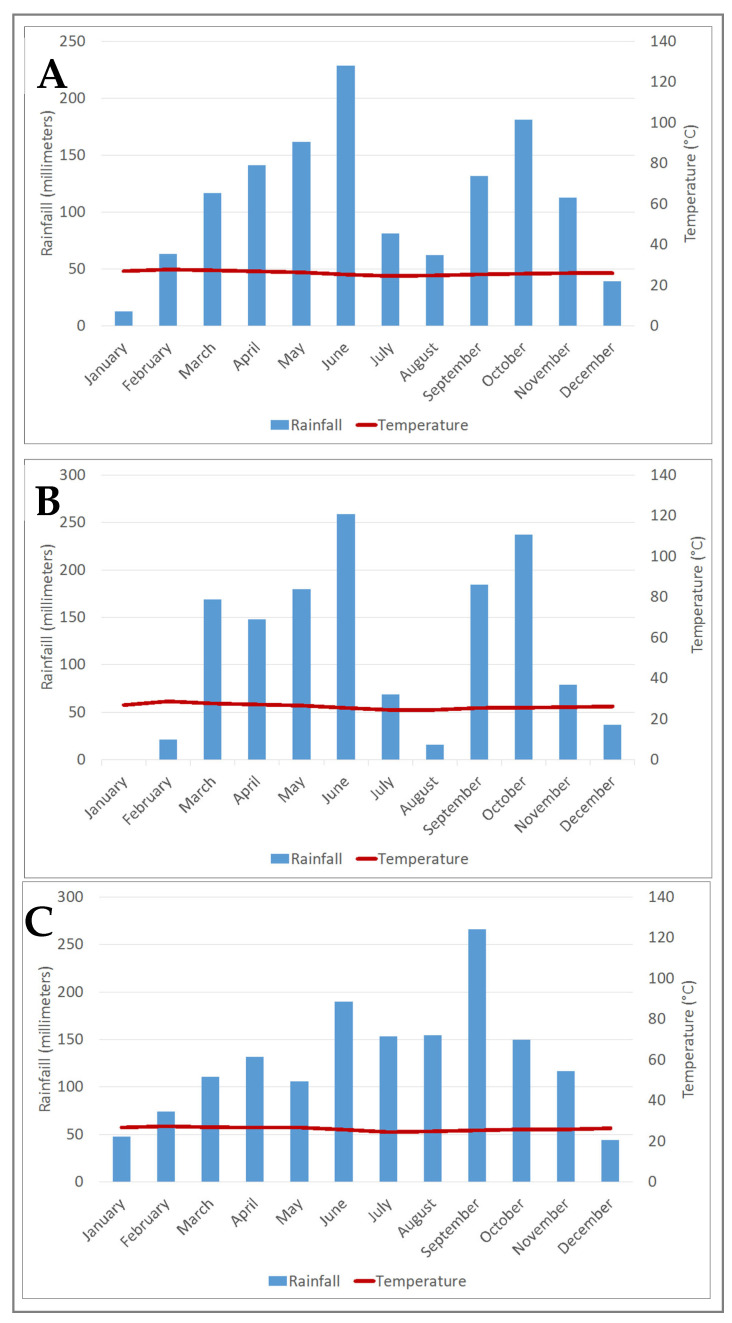
Ombrothermal diagram of the study area. (**A**): mean rainfall and temperature from 2012 to 2021, (**B**): rainfall and temperature in 2020, (**C**): rainfall and temperature in 2021.

**Figure 3 tropicalmed-10-00197-f003:**
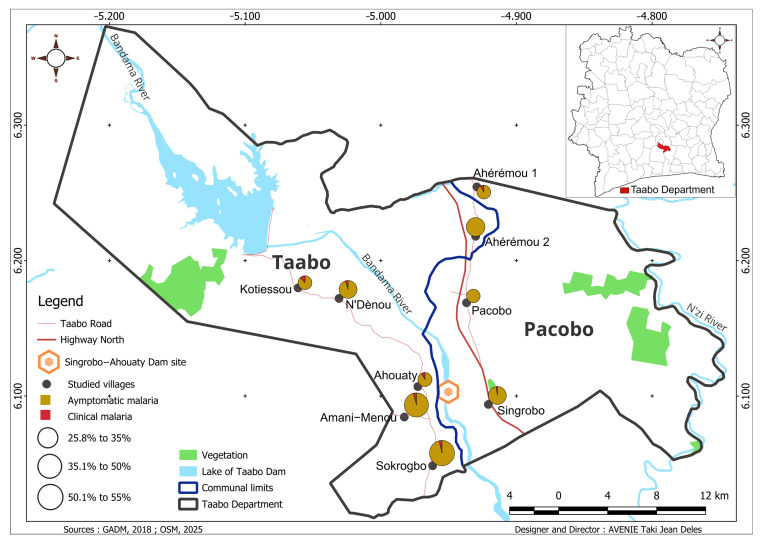
Map of the distribution of asymptomatic and symptomatic malaria infection in villages within the Singrobo–Ahouaty dam influence zone.

**Table 1 tropicalmed-10-00197-t001:** Malaria infection prevalence and parasite density by gender in the villages in 2021.

				Prevalence (%)			Parasite Density (Parasites/µL of Blood)					
Villages	Gender	N	*n*	Prevalence	OR (95% CI)	*p*	Parasite Density	Z	*p*	*P. falciparum*	*P. malariae*	Mix. Inf.
Ahérémou 1	Male	28	11	39.3	0.7 (0.2–2.2)	0.596	2650.4	−0.731	0.548	5	0	0
	Female	29	14	48.3			12,103.6			5	0	0
	Total	57	25	43.9			5663.9			10	0	0
Singrobo	Male	36	16	42.1	1.3 (0.5–3.0)	0.680	310.5	−0.265	0.812	10	0	0
	Female	67	26	36.6			302.3			20	0	0
	Total	103	42	38.5			305.0			30	0	0
Pacobo	Male	38	15	37.5	2.6 (0.9–7.6)	0.058	590.5	−0.840	0.414	8	0	0
	Female	53	10	18.5			1420.5			5	1	0
	Total	91	25	26.6			860.2			13	1	0
Ahouaty	Male	36	14	35.0	1.3 (0.5–3.6)	0.645	301.2	−0.579	0.573	10	0	0
	Female	40	13	29.5			459.2			8	0	0
	Total	76	27	32.1			363.3			18	0	0
Sokrogbo	Male	57	33	57.9	1.2 (0.6–2.5)	0.615	1208.2	−2.521	0.012	17	1	1
	Female	94	50	53.2			344.7			28	1	0
	Total	151	83	55.0			566.3			45	2	1
Ahérémou 2	Male	51	19	37.2	1.1 (0.5–2.4)	0.852	111.8	−0.990	0.336	8	0	0
	Female	72	27	35.1			163.8			7	0	0
	Total	123	46	35.9			133.6			15	0	0
Kotiessou	Male	29	13	44.8	1.2 (0.4–3.9)	0.797	116.6	−1.383	0.180	6	0	0
	Female	33	13	39.4			348.7			12	0	0
	Total	62	26	41.9			242.1			18	0	0
N’Dènou	Male	29	18	60.0	1.7 (0.6–4.8)	0.258	204.8	0.678	0.535	7	0	0
	Female	33	24	46.1			170.5			14	0	0
	Total	82	42	51.2			181.2			21	0	0
Amani–Menou	Male	61	36	59.0	1.5 (0.8–3.1)	0.242	608.3	−1.909	0.056	27	0	0
	Female	87	42	48.3			289.8			31	0	0
	Total	148	78	52.7			409.3			58	0	0
Global	Male	357	175	46.8	1.3 (1.0–1.7)	0.067	475.3	−1.528	0.127	98	1	1
	Female	518	219	40.5			359.8			130	2	0
	Total	875	394	43.1			405.7			228	3	1

N = number of individuals examined, *n* = number of individuals infected with malaria, CI = confidence interval, (Z, *p*) = non-parametric Mann–Whitney test.

**Table 2 tropicalmed-10-00197-t002:** Malaria infection prevalence and parasite density by age in the villages in 2021.

				Prevalence (%)			Parasite Density (Parasites/µL of Blood)					
Villages	Age Group	N	*n*	Prevalence	ꭓ^2^	*p*	Parasite Density	ꭓ^2^_kw_	*p*	*P. falciparum*	*P. malariae*	Mix. Inf.
Ahérémou 1	>14 years	34	8	23.5	16.168	<0.0001	7040.0	1.145	0.564	1	0	0
	6–13 years	14	12	85.7			2237.7			6	0	0
	0–5 years	9	5	55			33,747.5			3	0	0
Singrobo	>14 years	74	20	27.0	13.152	0.001	201.0	1.598	0.450	15	0	0
	6–13 years	9	5	55.6			261.8			5	0	0
	0–5 years	26	17	65.4			615.4			10	0	0
Pacobo	>14 years	57	12	21.1	2.532	0.282	416.8	1.680	0.432	5	1	0
	6–13 years	18	7	38.9			1673.1			6	0	0
	0–5 years	19	6	31.6			1027.6			2	0	0
Ahouaty	>14 years	48	9	18.8	10.739	0.005	228.6	0.990	0.610	6	0	0
	6–13 years	9	6	66.7			317.2			4	0	0
	0–5 years	27	12	44.4			550.4			8	0	0
Sokrogbo	>14 years	85	30	35.3	33.830	<0.0001	217.1	12.223	0.002	17	0	0
	6–13 years	29	27	93.1			520.9			15	0	0
	0–5 years	37	26	70.3			1696.4			13	2	0
Ahérémou 2	>14 years	96	28	29.2	14.843	0.001	110.3	2.049	0.359	10	0	0
	6–13 years	10	9	90.0			225.4			4	0	0
	0–5 years	22	9	40.9			112.0			1	0	0
Kotiessou	>14 years	35	10	28.6	5.903	0.052	126.6	1.631	0.442	9	0	0
	6–13 years	12	7	58.3			256.1			4	0	0
	0–5 years	15	9	60.0			742.8			5	0	0
N’Dènou	>14 years	53	20	37.7	11.965	0.003	90.2	3.153	0.207	9	0	0
	6–13 years	18	15	83.3			202.8			9	0	0
	0–5 years	11	7	63.6			1048.3			3	0	0
Amani–Menou	>14 years	85	34	40.0	13.435	0.001	188.3	18.242	<0.0001	24	0	0
	6–13 years	21	16	76.2			5227.4			14	0	0
	0–5 years	42	28	66.7			9361.2			20	0	0
Global	>14 years	567	171	30.2	111.155	<0.0001	160.4	43.383	<0.0001	96	1	0
	6–13 years	140	104	74.3			509.0			67	0	0
	0–5 years	208	119	57.2			1219.0			65	2	1

N = number examined, *n* = number infected with malaria, Mix. inf. = mixed infection, (Z, *p*) = non-parametric Kruskal–Wallis test.

**Table 3 tropicalmed-10-00197-t003:** Risk factors promoting malaria transmission in the Singrobo–Ahouaty dam influence zone in 2021.

Risk Factors	Modalities	N	*n* (%)	ORa (IC 95%)	*p*
Ocupation	#No schooling	11	5 (45.5)	NA	0.182
	Public sector employee	103	31 (30.1)	3.0 (0.8–11.5)	0.107
	Informal sector	70	20 (28.6)	1.5 (0.7–3.4)	0.017 *
	Raw material sector	10	2 (20.0)	1.4 (0.6–3.3)	0.048 *
	Market gardening	50	11 (22.0)	0.9 (0.2–4.8)	0.043 *
Monthly income	#60,000 FCFA and more	51	9 (17.7)	NA	0.074
	Less than 60,000 FCFA	134	46 (34.2)	0.7 (0.3–1.8)	0.037 *
	No income	59	14 (23.7)	1.7 (0.8–3.4)	0.216
Mosquito nuisance frequency	#Rare	44	7 (15.9)	0.4 (0.2–1.0)	0.017 *
	Several times a day	200	62 (31.0)		

N = number of people interviewed, *n* = number of people infected by malaria, % = proportion of people infected by malaria, ORa = adjusted odds ratio, IC95% = 95% confidence interval, NA = not applicable, (*) significant variables, and (#) reference category.

## Data Availability

The data used to write up the results leading to the unambiguous conclusions of this article are included in the article. The raw databases used for the statistical analyses of this manuscript are available on reasonable request.

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
