# Peer review of "Epidemiological Profile and Risk Factors for Malaria in Rural Communities Before the Operationalization of the Singrobo–Ahouaty Dam, Southern Côte d’Ivoire"

_tropicalmed, 2025, doi:10.3390/tropicalmed10070197_

Round 1

Reviewer 1 Report

Comments and Suggestions for Authors

This article addresses an important topic regarding epidemiological profile and risk factors of malaria in Côte d’Ivoire.  Methods are appropriate, results are robust and the manuscript well documented. However, some minor improvements to the manuscript are needed.

There are some typing errors across the document like “déadliest”, “finger prick”, “public dector”, etc…

The fourth paragraph of the introduction needs to be slightly restructured as it contains several repetitions:

  • “hydroelectric dams…can promote de spread of vector-borne diseases”
  • “hydroelectric dams have a major impact on the transmission vector-borne diseases such as malaria…”
  • “hydroelectric dams…by altering aquatic and environmental ecosystems…”
  • “the Singrobo dam could alter the local ecosystem…”

In the first paragraph of the “material and methods” section, it is not clear if the dam was already under construction at the time of the study (2021) or not. Should this study serve as a starting point for comparing the data collected before the dam is built with those collected after? In the same way, the discussion part is mainly focused on impact of gender, age, location and occupation, but the importance of the dam construction is not so well discussed.

In Figure 2, units for “precipitation” and “Temperature” are missing both in the axis titles and legend. Does the temperature unit °C or °F? (Values up to 140 are unrealistic for Celsius…).

In page 9, the paragraph “Malaria infection prevalence by age groups” is duplicated.

In tables 1 and 2, please adjust the character size so that “prevalence” appears on a line of its own.

Do the studies carried out in 2010 and 2011 concern the same geographical area? Does the dam construction have an impact on the epidemiology and risk factors of malaria in the area?

First line of page 17: “Overall, children under five years of age 0-5 years.” is not a sentence.

If some data are available regarding HRP2/3 deletion in Côte d’Ivoire, please add a comment in the discussion part.

Author Response

Comments 1: The English is fine and does not require any improvement.

Response 1: Thank you for your positive feedback regarding the quality of the English. We appreciate your comment.

Reviewer 2 Report

Comments and Suggestions for Authors

This manuscript examined factors that contribute to the risk of malarial incidences due to the development of dams in Côte d'Ivoire.
This was done through questionnaires and examination of bloodsmears for malarial parasites.

Future research should also include the presence of the vectors and the parasite in the vectors.

Expand on what species are present in Côte d'Ivoire (note reference 16 do not mention any vectors).

Page1:
Background 
Malaria remains a parasitic disease that continues to pose a significant public health threat worldwide, with the highest burden observed in sub-Saharan Africa [1–4].
The African region of the World Health Organization (WHO) bears the highest malaria burden, accounting for 94% of global cases (over 247 million cases) and 95% of global malaria-related deaths (over 567,000 deaths) [2].

Page 5:

The region’s environmental conditions, including high humidity levels and extensive areas, create permanent reproduction sites for malaria vectors (Anopheles gambiae, An. funestus and An. nili) such as malaria parasites (P. falciparum, P. malariae and P. ovale) [16].  - This reference (16) does not state anywhere which Anopheles species occur in Côte d'Ivoire. 

Look at the following references and add to the manuscript regarding species in Côte d'Ivoire:
N’Dri, B.P., Wipf, N.C., Saric, J. et al. Species composition and insecticide resistance in malaria vectors in Ellibou, southern Côte d’Ivoire and first finding of Anopheles arabiensis in Côte d’Ivoire. Malar J 22, 93 (2023). https://doi.org/10.1186/s12936-023-04456-y

Fournet, F., Adja, A.M., Adou, K.A. et al. First detection of the malaria vector Anopheles arabiensis in Côte d’Ivoire: urbanization in question. Malar J 21, 275 (2022). https://doi.org/10.1186/s12936-022-04295-3

Kouamé JKI, Edi CVA, Zahouli JBZ, Kouamé RMA, Kacou YAK, et al. (2024) Assessing species composition and insecticide resistance of Anopheles gambiae complex members in three coastal health districts of Côte d’Ivoire. PLOS ONE 19(12): e0297604. https://doi.org/10.1371/journal.pone.0297604

Yokoly FN, Zahouli JBZ, Small G, Ouattara AF, Opoku M, de Souza DK, Koudou BG. Assessing Anopheles vector species diversity and transmission of malaria in four health districts along the borders of Côte d'Ivoire. Malar J. 2021 Oct 18;20(1):409. doi: 10.1186/s12936-021-03938-1. PMID: 34663359; PMCID: PMC8524949

Assouho KF, Adja AM, Guindo-Coulibaly N, Tia E, Kouadio AMN, Zoh DD, Koné M, Kessé N, Koffi B, Sagna AB, Poinsignon A, Yapi A. Vectorial Transmission of Malaria in Major Districts of Côte d'Ivoire. J Med Entomol. 2020 May 4;57(3):908-914. doi: 10.1093/jme/tjz207. PMID: 31785095.

Add space between: 2020 and2021 (These conditions are further exacerbated by substantial annual rainfall, which reached 1398 millimeters and 1544 mm in 2020 and2021,

Study design:
This section is confusing as the 1st sentence indicates that this study focused on children but later on it shows that the study was done on a cross-section of the people (older and younger than 18 years) - clarify.

Page 9 and 14 
Duplication of last paragraph starting on page 9 (the Malaria infection prevalence by age groups paragraph) 

Table 3 and questionnaire (Table S1):
Correct: 
Ocupation to Occupation
Public dector employe to Public sector employee
Raw maternal sector to Raw material sector
Questionnaire (Table S1) this is both in English and French – use only 1 language:
Educational statut to Educational status
Causes of malaria: watercouses to watercourses 
soins de soins cas de maledie translate to care in case of illness
Self-medication: origine of care to origin of care
#Pharmacy on the floor? What does this mean? 
#Lost well? What does this mean?
Lutte antivectorielle intérieur to Indoor vector control
#Timor? What does this mean? Do you mean timer

Page 17:
Add also i.e: This could be explained by the nature of these activities, which are often carried out outdoors in areas close to wetlands or stagnant water, which are conducive to the proliferation of malaria vectors, particularly An. gambiae s.l. [49,50], which is also the main malaria vector in the study area.
Several other studies carried out in Côte d'Ivoire have also observed this phenomenon regarding income levels [45,51,52].

Comments on the Quality of English Language

Some language changes needed - see comments to authors.

Author Response

Comments 1: The English could be improved to more clearly express the research.

Response 1: Thank you for pointing this out. We agree with this comment. The English language of our manuscript has been thoroughly revised. We used the MDPI Author Services, specifically under the Standard Language Editing service, for professional editing of the entire document (Abstract to Conclusion). The authors have also revised the rest of the manuscript. We trust that the current version meets the required language standards for clear and effective communication of our research.